# Nonlinear Modeling of Contact Stress Distribution in Thin Plate Substrates Subjected to Aspect Ratio

**DOI:** 10.3390/s23084050

**Published:** 2023-04-17

**Authors:** Chao Lv, Huixin Wei, Zhiwen Lan, Ping Wu

**Affiliations:** Institute of Engineering Mechanics, Nanchang University, Nanchang 330031, China

**Keywords:** concentrated load, contact stress, aspect ratio, exponential function model

## Abstract

The foundation substrate’s basal contact stresses are typically thought to have a linear distribution, although the actual form is nonlinear. Basal contact stress in thin plates is experimentally measured using a thin film pressure distribution system. This study examines the nonlinear distribution law of basal contact stresses in thin plates with various aspect ratios under concentrated loading, and it establishes a model for the distribution of contact stresses in thin plates using an exponential function that accounts for aspect ratio coefficients. The outcomes demonstrate that the thin plate’s aspect ratio significantly affects how the substrate contact stress is distributed during concentrated loading. The contact stresses in the thin plate’s base exhibit significant nonlinearity when the aspect ratio of the test thin plate is greater than 6~8. The aspect ratio coefficient-added exponential function model can better optimize the strength and stiffness calculations of the base substrate and more accurately describe the actual distribution of contact stresses in the base of the thin plate compared to linear and parabolic functions. The correctness of the exponential function model is confirmed by the film pressure distribution measurement system that directly measures the contact stress at the base of the thin plate, providing a more accurate nonlinear load input for the calculation of the internal force of the base thin plate.

## 1. Introduction

When establishing power transmission and communication towers on soft soil near rivers and lakes, a flexible foundation slab is a common foundation form [1,2]. The form of foundation reaction force distribution greatly influences the internal force response of the foundation base plate. In engineering, a concentrated load is a common form of load used between the foundation and building, and the contact stress distribution between the thin slab and substrate with different width-to-height ratios is usually assumed to be linear [3,4,5]. However, actual contact stresses are generally nonlinearly distributed and can be saddle-shaped, parabolic, or anti-parabolic, which is mainly determined by the substrate bed factor or soil type and the material properties of the sheet [6]. The three most common nonlinear contact stress distributions described in the generalized Winkel foundation model are saddle-shaped, parabolic, and anti-parabolic [7,8]. Scholars have conducted studies on the contact stresses of foundation footings under concentrated loads, and Wang et al. [9] found that the measured field test values of foundation contact stress distribution present a convex parabolic form with small, middle, and large edges and the distribution of reaction forces does not vary much for different soils. Similarly, a large number of field measurements of foundation contact stress distribution in engineering practice show a similar parabolic variation law [10,11,12]. To account for this nonlinear variation law, Wang et al. [13] used a parabolic surface function to describe the foundation contact stress distribution and used K to control the adjustment function, where K > 1 for sandy foundations, K < 1 for clay foundations, and K = 1 for uniform distribution of reaction forces. Let the expression for the contact stress at a specific location in the foundation be represented by P_x=x_0__ = K · P_0_, where K is the control adjustment coefficient function. The relationship of K can be determined through experimental simulations. However, this parabolic function does not accurately reflect the nonlinear law of the stresses in the thin plate and the substrate, and a more accurate functional model is needed to characterize it.

The Technical Provisions for the Design of Power Transmission Line Foundations (DL/T5219-2005) [14] and the Code for the Design of Building Foundations (GB50007-2002) [15] assume linear contact stress distribution when the aspect ratio of foundation steps is less than or equal to 2.5 to simplify calculations. However, when the step width-to-height ratio exceeds 2.5, no calculation method for the internal force of the thin plate is provided in the specification. Determining the distribution of contact stresses between thin slabs and footings for calculating internal forces in foundation footings when the aspect ratio is greater than 2.5 is crucial for optimizing thin slab designs and reducing engineering costs [16]. The problem of contact stress distribution between the thin plate and the substrate has been studied by many scholars, and various methods have been developed to solve this problem. Solving the contact stress distribution of a four-sided free plate on an elastic substrate is important for calculating the analytical solution of displacement and internal forces in the plate [17]. Numerous scholars at home and abroad have studied the methods for solving the bending problem of a rectangular thin plate with four free sides on an elastic foundation, including the numerical method, analytical method, and semi-analytical method with semi-numerical values [18,19,20]. Each solution method must satisfy not only the fourth-order differential bending equation for the plate [21] but also the geometric and internal force boundary conditions on the four free edges [22]. However, the geometric and internal force boundary conditions related to the form of distribution of contact stresses [23,24] are essential for the bending equations [25]. Therefore, accurately describing the distribution form of contact stresses through model studies is necessary. In engineering and scientific fields, measuring contact stress is vital for evaluating material performance, designing new products, and optimizing production processes.

Many mechanical systems’ durability and performance are significantly impacted by contact stress, a crucial factor. The deformation, wear, and failure of these parts can all be significantly influenced by the distribution of stresses that take place at the interface between two contacting bodies. Therefore, for designing and optimizing mechanical systems, it is crucial to comprehend how contact stress arises and how it can be measured and controlled. Guan investigated the creation of tri-axial stress measuring and sensing technology for tire-pavement contact surfaces [26] and described a cutting-edge method for determining the tri-axial stress distribution at the tire–pavement contact surface using a sensor array [27].

First of all, it has been hard to quantify contact stress. Secondly, the distribution of contact stress for thin plates is primarily linear and parabolic, which is relatively conservative for the input conditions of internal force calculation. In this paper, we introduce an innovative type of nonlinear distribution of contact stress called the exponential function model. In order to determine contact stresses in the substrate of thin plates with different aspect ratios, this paper performs film pressure measurement tests. This method takes into account the problems posed by the aspect ratio of the base plate as well as the phenomenon of nonlinear stress distribution in thin plates and substrates. We investigate how aspect ratio affects contact stress distribution and propose a nonlinear distribution model of contact stresses in the substrate of thin plates under concentrated loading. The results of our study have important theoretical and engineering value by providing more precise external boundary conditions for designing and calculating the strength and stiffness of thin plates with different aspect ratios.

## 2. Experimental Programs

This section first introduces the component composition and working principle of the thin film pressure distribution test system, as well as the data acquisition method and process. Then, an experimental plan is developed to obtain the real contact stress distribution of the thin plate substrate under graded loading by changing the aspect ratio of the thin plate used in the test and based on the test of the thin film pressure distribution measurement system.

### 2.1. Introduction to Thin Film Pressure Distribution Measurement System

The film pressure distribution measurement system is divided into a hardware part and a software part [28]. The hardware part consists of a polyester film sensor, an Analog to Digital Converter (A/D) conversion circuit for the sensor (handle), and software based on a PC using C language to complete an intelligent display system on a microcontroller.

In the test device, Figure 1a shows a 5076-P1-35611DT1-50 type thin film pressure sensor. The area of the thin film pressure sensor is 83.8 mm wide × 83.8 mm high. The pressure sensor consists of two very thin polyester films, as shown in Figure 1b, where the inner surface of one film is laid with a number of band conductors arranged in rows, and the inner surface of the other film is laid with a number of band conductors arranged in columns conductors. The intersection of the rows and columns is the pressure sensing unit, which has a total of 44 rows and 44 columns, forming a uniform distribution of 1936 measurement points. The conductors are made of conductive material with a certain width, and the distance between rows can be changed according to requirements. Therefore, polyester film sensors are available in a variety of sizes and shapes and can be used in different contact conditions. The outer surface of the conductor in the sensing area is coated with a special pressure-sensitive semiconductor material. When two films are combined into one, the intersection of a large number of transverse and longitudinal conductors forms an array of stress-sensing points. When an external force is applied to the sensing unit, the resistance value of the conductor changes linearly with the change in the external force, thus reflecting the stress value at the sensing point. The resistance value is maximum when the pressure is zero and decreases as the pressure increases. This linear change in voltage can reflect the magnitude and distribution of the pressure between the two contact surfaces. The small thickness and high flexibility of the thin-film pressure sensor have no influence on the contact environment, so it can directly measure the real contact stress distribution under different conditions and has the characteristics of high measurement accuracy. 

The connection diagram between the conductive handle and the sensor is shown in Figure 1a. The connection part in the handle has a conductive interface that can transmit the electrical signal from the sensor film to the computer.

The handle is the connection device between the computer software and the sensor and is also the A/D converter. When an external force is applied to the sensing point, the resistance value of the semiconductor changes proportionally to the change in the external force. This change in electrical signal is transmitted to the control circuit on the handle, which is then input to the software and displayed on the computer screen to reflect the stress value and its distribution at the sensing point.

The width and spacing of the conductors within the sensor determine the number of sensing points per unit area and the spatial resolution, which can be determined as needed. The value of the spatial resolution on the sensor area can meet a variety of measurement requirements. The sensors can be fabricated in various dimensions and configurations, exhibiting a stress measurement capacity ranging from 0.1 to 175 MPa. The unique feature of this grid sensor is that the sensing area is completely insulated from the non-sensing area. Knowing the spatial size distribution of the sensing points, the force applied to the sensor on a certain area can be intelligently digitized and displayed with Light Emitting Diode (LED) or Liquid Crystal Display (LCD) using the software.

The software part of the thin film pressure distribution measurement system is used to process the measured 2D matrix voltage data and convert it into 2D graphics or 3D graphics display. When a calibration file is not added, the initial values are displayed without units. When the calibration file is added, the initial values are converted into force values and divided into 17 color levels to display the measured stress distribution and the magnitude of the combined force values, indicated from small to large, in a blue-to-pink gradient 2D display graphic. Each pixel point corresponds to the intersection of the band conductors in the film, where the color reflects the measured data at the level set by the corresponding calibration file. In addition, the calibration file is different, and the same force value will be displayed in different colors.

### 2.2. Experimental Program

Square glass sheets of the same width and different thicknesses are used as test specimens, and the widths of the specimens are all 75 mm, and the thicknesses are 4 mm, 5 mm, 8 mm, 10 mm, and 12 mm, respectively. Finally, the test molds with aspect ratios of 18.75, 15, 9.375, 7.5, and 6.25 are obtained, as shown in Figure 2a,b.

The actual stress distribution of the square plate when subjected to the concentrated load is observed by applying a graded concentrated load on the center of the square glass plate and placing a thin-film pressure transducer at the bottom. The width of the glass plate is kept constant in the test, and the thickness of the glass is varied to achieve a change in the aspect ratio. A schematic diagram of the test assembly is shown in Figure 3 below. The overall picture of the experimental assembly can be visualized according to Figure 3.

The loading device for the test is a microcomputer-controlled electronic universal testing machine, as shown in Figure 4. The loading speed should not be too fast to avoid irregular discrete damage by too rapid force, and 10 N/s is used. Glass plates with different aspect ratios are placed on the upper part of the pressure film and loaded by 250 N, 500 N, 750 N, 1000 N, and 1250 N in a graded manner. The film pressure distribution system is used to measure the real contact stress distribution of the thin plate substrate, and finally, the system-recorded data is displayed.

## 3. Test Results and Analysis

The film pressure distribution measurement system displays the combined value of contact stresses in thin plates with different aspect ratios under graded concentrated loads measured by 1936 measurement points on the film pressure sensor. The actual distribution of the contact stress of the thin plate is also reflected visually using a cloud map. By repeating the test several times, the accuracy of the film pressure distribution measurement system and the real situation of the contact stress distribution in the substrate of the thin plate can be verified by comparing the combined values of the load and the test results. The test results provide a basis for a more accurate study of the contact stress distribution at the base of the thin plate.

### 3.1. Combined Force Value of Film Pressure Distribution Measurement Results

By applying graded concentrated loads (250 N, 500 N, 750 N, 1000 N, and 1250 N) to square thin plates with different aspect ratios (18.75, 15, 9.375, 7.5, and 6.25), the stability of the combined force values is demonstrated in the test results. The reproducibility of the test has an important impact on the use of the film pressure distribution measurement system, and for this reason, multiple sets of data are measured for each load in the contact stress distribution test on thin plates with different aspect ratios. Studying whether the measured combined force value of the film pressure distribution measurement system is controlled within a certain error range under the same electronic universal testing machine load provides an accurate and realistic distribution for the study of contact stress distribution.

The following conclusions can be obtained from Table 1. First, the film pressure distribution measurement system can basically control the input-graded load value and the output stress combined force value within an error range of 10%. When measuring the contact stress, the total combined force in the test result can be lower than the applied load, resulting in a negative test error. Second, for the thin plate with a large aspect ratio, the combined force value measured by the test has a relatively large error in comparison with the load. The reason for this is that when the concentrated load reaches 1000 N or more, the stress concentration phenomenon may occur, or the stress value at the central position of the sensor exceeds the range of its measuring point due to the large aspect ratio of the thin plate. Thirdly, as the aspect ratio increases, the combined force value of the test results and the load are gradually matched, and the error can be controlled even within 5%, which provides an accurate test basis for the study of contact stress distribution. The overall test results show that the contact stress measurements can be controlled within a reasonable range regardless of the aspect ratio, and the test results of the film pressure measurement system are relatively reliable [29].

### 3.2. Characteristics of Nonlinear Distribution of Contact Stress in Thin Plate Substrate

The results of the contact stress test measurements at the bottom of the thin plate can be represented visually in the measurement software by means of a stress cloud [30]. The magnitude of the contact stress is differentiated by color, and the stress is reflected as a gradation from blue to pink in the cloud.

As an example, in the test results for a square glass plate with a maximum aspect ratio of 18.75, the distribution of substrate contact stresses exhibited a significant nonlinearity when the aspect ratio of the sheet is fixed and different loads are applied. Figure 5a–e show the distribution clouds of contact stresses under concentrated loads of 250 N, 500 N, 750 N, 1000 N, and 1250 N, respectively.

From Figure 5, it can be seen that for the same square glass sheet under the central concentrated load, the total contact area of the sheet and the substrate is fixed. When the concentrated load is small, the pink area with larger contact stress in Figure 5a,b is small in proportion to the total contact area. With the increase of load, the corresponding area of contact stress increases, and the area with larger contact stress also increases gradually, which is shown by the expansion of the pink area in Figure 5d,e.

Similarly, when the thin plate load is fixed and the aspect ratio of the plate is changed, the test results of the basal contact stress distribution are compared, and it is found that there is also a significant difference in the form of basal contact stress distribution. Examining the distribution of basal contact stresses in thin plates with different aspect ratios under the same size of the concentrated load, as shown in Figure 6. Figure 6a–e show the distribution clouds of contact stresses for thin plates with aspect ratios of 18.75, 15, 9.375, 7.5, and 6.25 under a fixed load of 1000 N, respectively.

From Figure 6a,b, we can see that when the aspect ratio of the plate is changed under a fixed load, the pink area in the contact stress distribution of the thin plate with aspect ratios of 18.75 and 15 is larger and non-linearly obvious. When the aspect ratios are 9.375, 7.5, and 6.25, respectively, which are shown in Figure 6c–e, the color difference of the stress cloud becomes smaller. As the aspect ratio of the plate decreases, the contact stress distribution gradually tends to be uniform, indicating that the aspect ratio of the plate has a significant effect on its contact stress distribution.

## 4. Modeling of Exponential Function Distribution of Contact Stress under Concentrated Load

An analytical model of the contact stress in the base of the thin plate is established, as shown in Figure 7, where a is the side length of the square thin plate, δ is the thickness of the square thin plate, and the aspect ratio is defined as λ=a/δ. A concentrated load *F* is applied at point *O* (*a*/2, *a*/2) in the center of the thin plate. The x-axis and y-axis represent the plane where the test plate is located, and the z-axis indicates the direction of the applied load and the direction of the contact stress.

In the introduction, Wang [13] mentioned that parabolic functions can be used to reflect the distribution form of contact stresses in the base slab, and the contact stress distribution function at the bottom of the slab is assumed to be
(1)$$p\left(x,y\right)=A_{1}x^{2}+A_{2}y^{2}+A_{3}x+A_{4}y+A_{5}$$

The five parameters *A1*, *A2*, *A3*, *A4*, and *A5* in Equation (1) are coefficients to be determined, which can be determined by a stress balance equation and a continuous condition of the reaction force distribution at four corner points. However, the process of solving is complicated. In addition, in the actual situation, the distribution form of contact stress is not one kind of paraboloid, and there are other common nonlinear distribution cases, such as saddle plane and bell shape. In order to better describe the nonlinear distribution of contact stresses in the base of a square thin plate with four free sides under concentrated loading, an exponential function contact stress distribution model is proposed in this paper [31].
(2)$$p(x,y)=A+Be^{C\cdot \frac{{\left(x-a/2\right)}^{2}+{\left(y-a/2\right)}^{2}}{{\left(a/2\right)}^{2}}}$$

Equation (2) can simultaneously describe a variety of nonlinear contact stress distribution forms, including parabolic, saddle surface, and bell-shaped, among other nonlinear shapes, where *A*, *B*, and *C* are coefficients to be determined.

The magnitude of the contact stress in the center of the thin plate is defined as the average force under the plate multiplied by the corresponding aspect ratio coefficient *p*, where *p* = *kp*_0_, *p*_0_ = *F*/*a*^2^. *p* and *p*_0_ are the actual central contact stress of the thin plate and the uniform force of the thin plate, respectively. “*k* = *k*(*λ*)” is the aspect ratio coefficient of the thin plate. According to the equilibrium equation, the symmetry of the square thin plate structure and the central contact stress can determine the coefficients *A*, *B*, and *C* of Equation (2).

According to the previous definition, the contact stress at the center of the thin plate is the mean contact stress multiplied by the aspect ratio factor:(3)$$p(\frac{a}{2},\frac{a}{2})=\frac{kF}{a^{2}}$$

Integration along the region leads to the equilibrium equation:(4)$$\int_{0}^{a}\int_{0}^{a}p(x,y)dxdy=F$$

For a square thin plate, the concentrated load acts at the center of the plate, and the square thin plate structure are symmetrical. Therefore, the contact stress distribution model of the substrate is also symmetrical.
(5)p(x,y)=p(y,x)

The corresponding coefficients to be determined can be obtained by solving the system of equations from (3) to (5).

Different external conditions may have an effect on the initial contact stress value of the contact stress at the base of the thin plate, but under the test conditions, the environment is stable, the 4 sides are free, and the boundary contact stress value is 0. In order to make the model meet the external conditions when building the model, the value of *A* is, therefore, taken as 0.
(6)A=0

Substituting the center coordinates (*a*/2, *a*/2) in Equation (2), Equation (7) is obtained.
(7)$$B=\frac{kF}{a^{2}}$$

The values of *A* and *B* that have been determined are substituted into the contact stress distribution model (2) and integrated along the region to obtain Equation (8). *F* is calculated as follows:(8)$$\int_{0}^{a}\int_{0}^{a}p(x,y)dxdy=\int_{0}^{a}\int_{0}^{a}\frac{kF}{a^{2}}e^{C\cdot \frac{{\left(x-a/2\right)}^{2}+{\left(y-a/2\right)}^{2}}{{\left(a/2\right)}^{2}}}dxdy=F$$

Using the polar coordinate transformation, let $\left\{ \begin{array}{c} x=r\cos\theta +a/2\\ y=r\sin\theta +a/2 \end{array}\right.$, where the integration regions of a/2=rcosθ, r∈ [0,a/2cosθ],θ∈ [0,π/4] coincide exactly with the original integration region (the integration region is shown in Figure 8). Equation (9) is obtained. The transformation of the integration region can be better understood in Figure 8.
(9)$$4\int_{0}^{\pi /4}d\theta \int_{0}^{a/2\cos\theta }\frac{kF}{a^{2}}e^{C\cdot \frac{r^{2}}{{\left(a/2\right)}^{2}}}rdr+4\int_{\pi /4}^{\pi /2}d\theta \int_{0}^{a/2\sin\theta }\frac{kF}{a^{2}}e^{C\cdot \frac{r^{2}}{{\left(a/2\right)}^{2}}}rdr=F$$

The Procedure for calculating Equation (9) is as follows:(10)$$4\int_{0}^{\pi /4}d\theta \int_{0}^{a/2\cos\theta }\frac{kF}{a^{2}}e^{C\cdot \frac{r^{2}}{{\left(a/2\right)}^{2}}}\frac{{\left(a/2\right)}^{2}}{2C}d\frac{Cr^{2}}{{\left(a/2\right)}^{2}}+4\int_{\pi /4}^{\pi /2}d\theta \int_{0}^{a/2\sin\theta }\frac{kF}{a^{2}}e^{C\cdot \frac{r^{2}}{{\left(a/2\right)}^{2}}}\frac{{\left(a/2\right)}^{2}}{2C}d\frac{Cr^{2}}{{\left(a/2\right)}^{2}}=F$$
(11)$$\frac{kF}{2C}\left(\int_{0}^{\pi /4}\left(e^{C/\cos^{2}\theta }-1\right)d\theta +\int_{\pi /4}^{\pi /2}\left(e^{C/sin^{2}\theta }-1\right)d\theta \right)=F$$

The simplification yields Equation (11).
(12)$$4\int_{0}^{\pi /4}d\theta \int_{0}^{a/2\cos\theta }\frac{kF}{a^{2}}e^{C\cdot \frac{r^{2}}{{\left(a/2\right)}^{2}}}\frac{{\left(a/2\right)}^{2}}{2C}d\frac{Cr^{2}}{{\left(a/2\right)}^{2}}+4\int_{\pi /4}^{\pi /2}d\theta \int_{0}^{a/2\sin\theta }\frac{kF}{a^{2}}e^{C\cdot \frac{r^{2}}{{\left(a/2\right)}^{2}}}\frac{{\left(a/2\right)}^{2}}{2C}d\frac{Cr^{2}}{{\left(a/2\right)}^{2}}=F$$

Since the integration region and the integration function are the same as for $\int_{\pi /4}^{\pi /2}\left(e^{C/\sin^{2}\theta }\right)d\theta$ $\int_{0}^{\pi /4}\left(e^{C/\cos^{2}\theta }\right)d\theta$ satisfies $\theta^{\prime }=\theta +\pi /2$), Equation (13) is obtained.
(13)$$\begin{array}{l} \sin^{2}\theta =\frac{1-\cos2\theta }{2}\\ =\frac{1-[1-{\left(2\theta \right)}^{2}/2!+{\left(2\theta \right)}^{4}/4!+\ldots \ldots +{\left(2\theta \right)}^{n}/n!]}{2}\\ =\theta^{2}-\frac{4}{3}\theta^{4}+\ldots \ldots \end{array}$$

Equation (13) is approximately equal to Equation (14) and finds the solution of *C*.
(14)$$\int_{\pi /4}^{\pi /2}\left(e^{C/\theta^{2}}\right)d\theta =\int_{\pi /4}^{\pi /2}\left(1+C/\theta^{2}+C^{2}/2\theta^{4}+\ldots \ldots \right)d\theta =C/k+\pi /4\;C=3\pi^{2}\frac{\pi -2k}{28k}$$

Finally, the contact stress distribution model, including the aspect ratio factor for a square thin plate under a central concentrated load, is derived as Equation (15).
(15)$$p(x,y)=\frac{kF}{a^{2}}e^{3\pi^{2}\frac{\pi -2k}{28k}\cdot \frac{{\left(x-a/2\right)}^{2}+{\left(y-a/2\right)}^{2}}{{\left(a/2\right)}^{2}}}$$

## 5. Comparison of Model Results with Experimental Data

The experimental results of the film pressure distribution measurement system show that the contact stress distribution in the substrate of a square thin plate under concentrated loading is nonlinear. The aspect ratio of the plate has a significant effect on the contact stress distribution in the substrate. In this section, the correctness of the exponential function contact stress distribution model proposed in this paper is investigated by further analysis of the experimental results and comparison with the theoretical values of different contact stress distribution models.

### 5.1. Comparison of Model and Experimental Stress Values of Contact Stress Distribution by Exponential Function

In this section, the data analysis is processed according to the contact stress exponential function distribution model Equation (15) presented in Section 4 for thin plates with different aspect ratios. It is proved through experimental practice that the aspect ratio k proposed in this paper is generally between 2*λ*/15 and *λ*/5. The test results in Section 3 are compared with the model values of Equation (15) in a three-dimensional simulation. The applicability of the exponential function model proposed in this paper for nonlinear contact stress distribution can be visualized from the three-dimensional comparison graph.

Figure 9a–e are 3D comparison plots of the results comparing the theoretical and experimental values of the exponential function model under a concentrated load of 1000 N for aspect ratios *λ* of 18.75, 15, 9.375, 7.5, and 6.25, respectively. The *x*- and *y*-axes are the coordinates of the plane position of the sheet (in cm), the *z*-axis is the corresponding contact stress force value (in N/cm^2^), the scatter point is the stress value measured by the test induction unit, and the colored surface is the model result. It can be intuitively concluded that when the aspect ratio is above 6~8, the nonlinearity of the contact stress distribution is obvious, and the model of this paper has a high degree of fit to the test results. When the aspect ratio is below 6~8, the contact stress distribution tends to be uniform, and the nonlinearity is reduced.

In order to further analyze the model results, this paper uses the film pressure distribution measurement system under concentrated load to sense the distribution of measurement points as 44 × 44, with a total of 1936 measurement points. The amount of test data is large. In addition, due to the range of the thin film pressure sensor, some regions will have stress concentration, which does not play a better role in comparing the experimental results with the model results. On the contrary, choosing the central path comparison can reflect the nonlinearity of stress distribution well. The contact stress distribution of the square thin plate under concentrated loading is symmetrical, so it can also be extended to the contact stress distribution of the whole thin plate substrate, and the specific contact stress values are obtained by selecting the corresponding paths. In order to facilitate the comparison between the model stress values and the experimentally measured stress values, the central path of the surface equation in Equation (15), i.e., y = *a*/2, can be chosen to obtain the contact stress distribution function Equation (16) on this path when comparing the exponential function contact stress distribution model Equation (15) proposed in Section 4 of this paper with the experimental stress values.
(16)$$p(x,y)=\frac{kF}{a^{2}}e^{3\pi^{2}\frac{\pi -2k}{28k}\cdot \frac{{\left(x-a/2\right)}^{2}}{{\left(a/2\right)}^{2}}}$$

For the analysis and processing of a large number of experimental measurement data, the general value of the aspect ratio coefficient *k* is taken between 2*λ*/15 and *λ*/5. In this paper, we take *k* = *λ*/5, determine the value of *k*, and substitute it into Equation (16) to obtain Equation (17).
(17)$$p(x)=\frac{\lambda F}{5a^{2}}e^{3\pi^{2}\frac{5\pi -2\lambda }{28\lambda }\cdot \frac{{\left(x-a/2\right)}^{2}}{{\left(a/2\right)}^{2}}}$$

Equation (17) is the exponential function distribution model of contact stress for the central path of the thin plate under concentrated load containing the aspect ratio.

When the load is fixed and the aspect ratio of the thin plate is changed, the theoretical value curves of the exponential function contact stress distribution model and parabolic model of the thin plate are compared with the corresponding experimental values in Section 3. Finally, the results comparing the theoretical values of the model with the experimental values when the aspect ratio *λ* is 18.75, 15, 9.375, 7.5, and 6.25 are obtained, as shown in Figure 10.

From the comparison of the above model’s theoretical values and the corresponding test values, the following conclusions can be drawn:

1For the thin plate with an aspect ratio of 6~8 or more in Figure 9, the contact stress values of the tested bottom plate are nonlinearly distributed, and the exponential function contact stress distribution model is closer to the test results than the original parabolic model, which verifies the correctness of the model in the case of a larger aspect ratio;2For the thin plate with an aspect ratio below 6~8, it can be seen from Figure 9 that as the aspect ratio decreases, the contact stress value of the bottom plate is gradually stabilized within a certain stress range, and the exponential function contact stress distribution model and the parabolic model also converge at the same time. This phenomenon shows that when the aspect ratio is small, the exponential function contact stress distribution model and the original parabolic model are equally feasible and can reflect the contact stress distribution.

### 5.2. Error Analysis of Model and Test Values

The correctness of the exponential function contact stress distribution model can be verified by comparing the theoretical value curves of the exponential function contact stress distribution model on the central path of the thin plate in Section 5.1, the parabolic surface model, and the experimental results in Section 3. In order to apply the model to the engineering practice of foundation base plate design, an error analysis is performed for the above comparison results [32].

In the error analysis, the center of the thin plate is selected to be subjected to a concentrated load of 1000 N. A total of 5 intervals in the plate, marked as intervals 1, 2, 3, 4, and 5, are taken in the *x*-axis upward past the center. Their specific locations are [7.6~11.4 mm], [19~22.8 mm], [30.4,34.2 mm], [41.8~45.6 mm], [53.2~57 mm], and [53.2~57 mm] in the central path of Figure 11. The selection of the error interval can be visualized in Figure 11.

The average values of test stresses in these intervals are compared with the theoretical stress values of different models in order to avoid, as much as possible, the chance brought by the test measurements. The results of the error analysis are obtained, as shown in Table 2, Table 3, Table 4, Table 5 and Table 6.

As seen from Table 2, Table 3 and Table 4, for the thin plate with an aspect ratio greater than 8, the contact stresses at the base are calculated according to the assumption of linear distribution, and the contact stress values have a large error with the experimental values, with the error reaching 36.35~51.18%. The linear model is the most commonly used model in engineering today. It is conservative in design and does not take into account the non-linear nature of the contact stress distribution [33,34]. As a result, it has a larger error compared to the non-linear parabolic model and the exponential function model proposed in this paper. However, the exponential function contact stress distribution model has an error of within 5% between the stress value and the test stress value. It can be seen that the exponential function model can more accurately represent the contact stress distribution of the thin plate under concentrated loading.

From Table 5 and Table 6, for the thin plate with an aspect ratio below 8, the substrate contact stress distribution gradually tends to be uniform, and the errors between the exponential function contact stress distribution model and the test stress values are less than 5%. Meanwhile, the parabolic model values are compared with the experimental values, and there is an error of 38.18%~39.54%. The large parabolic model error under concentrated loading, when the aspect ratio is relatively large, is mainly because the model requires stress equations and reaction forces at the four corner points when determining the model coefficients [8]. However, the free boundary conditions under this test condition resulted in inaccurate reaction forces at the four corner points, which caused an increase in parabolic error. The linear model values used in engineering are on the conservative side, and there is an error of 20.03~24.03% compared with the experimental values. The reason for the small error is that the exponential function model proposed in this paper inherently includes aspect ratios, especially for thin plates with large aspect ratios, which are more consistent with the distribution of real substrate contact stresses. Thus, we can see that the exponential function contact stress distribution model proposed in this paper has the characteristics of small error and accurate calculation, which has certain practical significance in engineering applications.

## 6. Conclusions

Multiple measurement tests based on experimental measurements of the film pressure distribution measurement system are used to confirm the repeatability of the system. For measuring the base cohesion values and contact stress distribution forms of thin plates with various aspect ratios, the multiple tests also provide precise experimental support. The contact stress distribution can be studied more thoroughly, and the impact of aspect ratio on it can be confirmed by controlling the base force value within a tolerable error range. A graded loading test scheme is created for various load forms based on the thin film pressure distribution measurement system. The thin film pressure transducer directly measures the real contact stresses in the substrate of the sheet with various aspect ratios and material properties to produce the nonlinear situation of the real contact stress distribution.

The form of the contact stress distribution at the bottom of thin plates with various aspect ratios is examined in this paper, and a model for the contact stress distribution with an exponential function and aspect ratio coefficients is suggested. The model, which can more accurately describe the nonlinear situation of the actual distribution of contact stresses at the bottom of the substrate and is supported by the experimental data, has a straightforward form and only requires the determination of an aspect ratio coefficient. The theoretical value of the exponential function model, the experimentally measured value of the real contact stress, and the nonlinear model of the current contact stress distribution are compared. Additionally, the corresponding error analysis is derived to confirm the accuracy of the model in this paper. The exponential function distribution model of contact stress, which is more accurate and consistent with the actual nonlinear distribution, is proposed in accordance with the characteristics of the real test values of the contact stress distribution and model distribution. The exponential function contact stress distribution model can provide nonlinear load input and more precise external force boundary conditions for the internal force calculation of thin slabs with aspect ratios of 6~8 or more, improving the strength and stiffness calculation of foundation base slabs. This model is more accurate than the original linear model of contact stress distribution and the parabolic model.

Only the contact stress distribution in the base of thin plates with different aspect ratios under concentrated loading is taken into account by the exponential function contact stress distribution model proposed in this paper. However, the subsequent work will look into the thin plate’s material characteristics as well as additional influencing factors like the type of loading action.

## Figures and Tables

**Figure 1 sensors-23-04050-f001:**
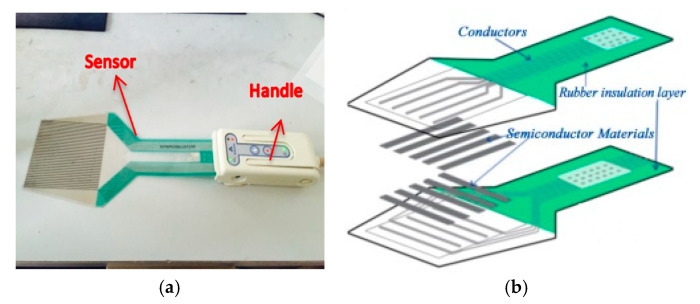
Thin film pressure sensor: (**a**) Tekscan sensor and (**b**) sensor composition.

**Figure 2 sensors-23-04050-f002:**
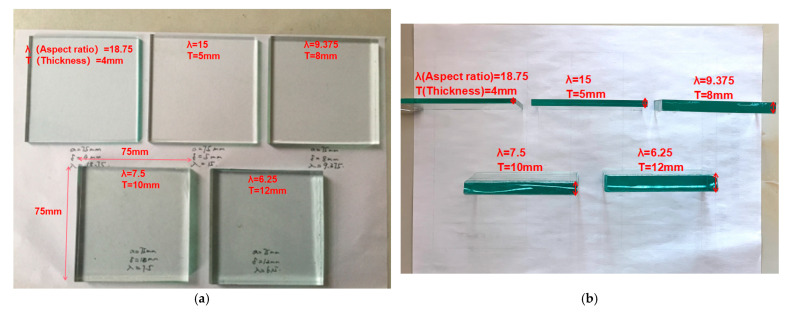
Glass sheet test pieces with different aspect ratios: (**a**) Top view; (**b**) Side view.

**Figure 3 sensors-23-04050-f003:**
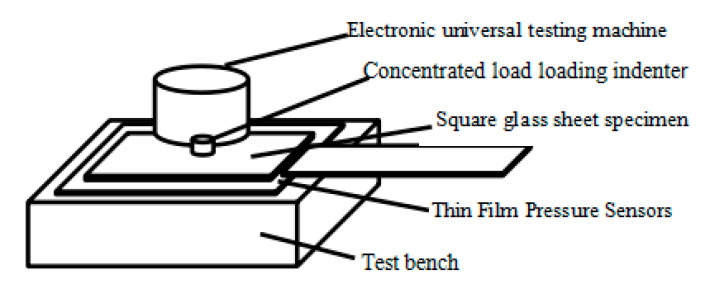
The schematic diagram of the test assembly.

**Figure 4 sensors-23-04050-f004:**
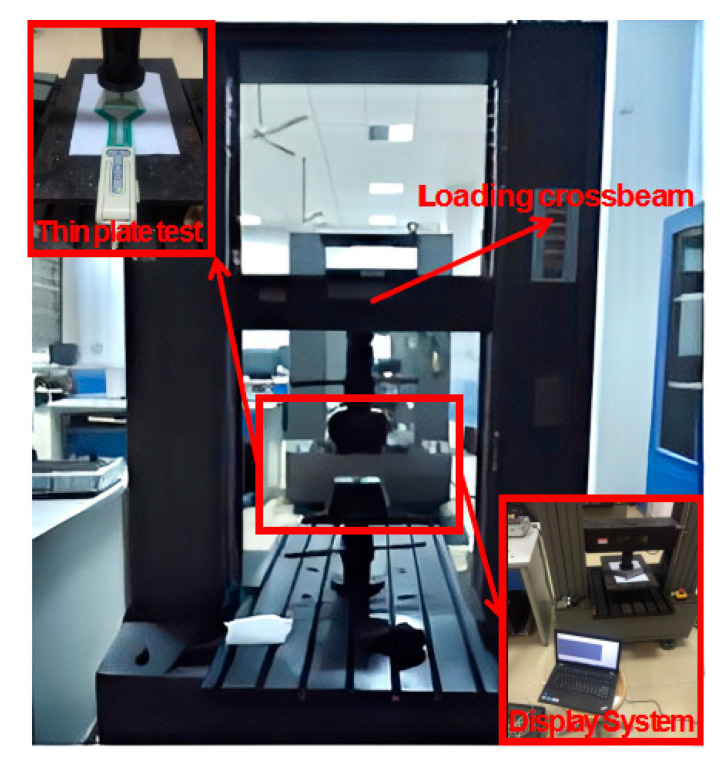
The loading device.

**Figure 5 sensors-23-04050-f005:**
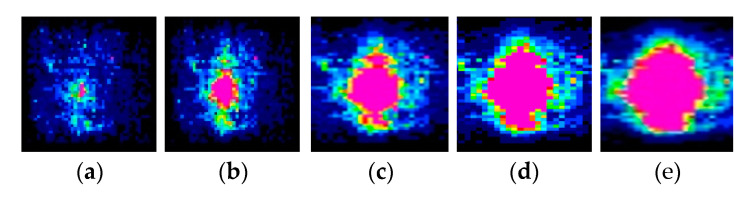
The distribution of contact stress distribution of thin plate with the change of load distribution: (**a**) load of 250 N; (**b**) load of 500 N; (**c**) load of 750 N; (**d**) load of 1000 N; and (**e**) load of 1250 N.

**Figure 6 sensors-23-04050-f006:**
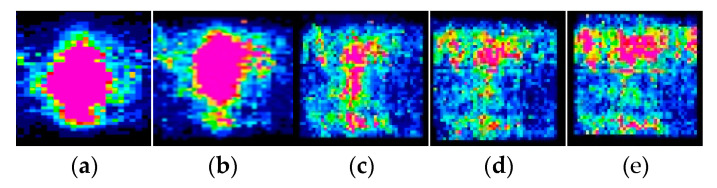
The distribution of contact stress distribution along with the change of the width and height of the thin plate: (**a**) aspect ratio of 18.75; (**b**) aspect ratio of 15; (**c**) aspect ratio of 9.375; (**d**) aspect ratio of 7.5; and (**e**) aspect ratio of 6.25.

**Figure 7 sensors-23-04050-f007:**
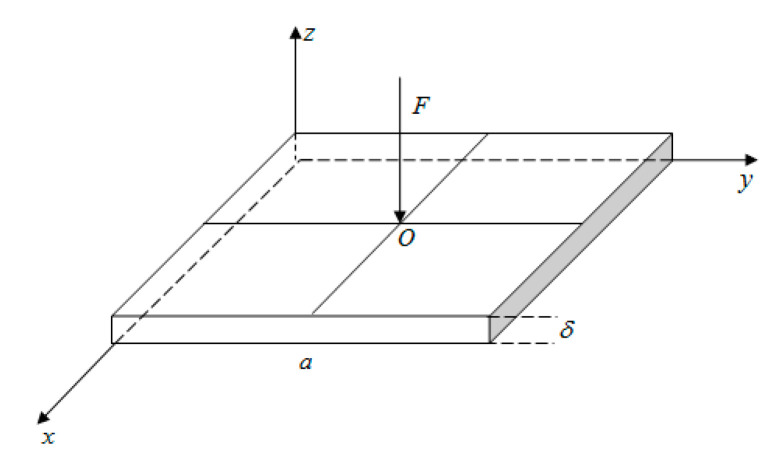
The analysis model of plate contact stress distribution function.

**Figure 8 sensors-23-04050-f008:**
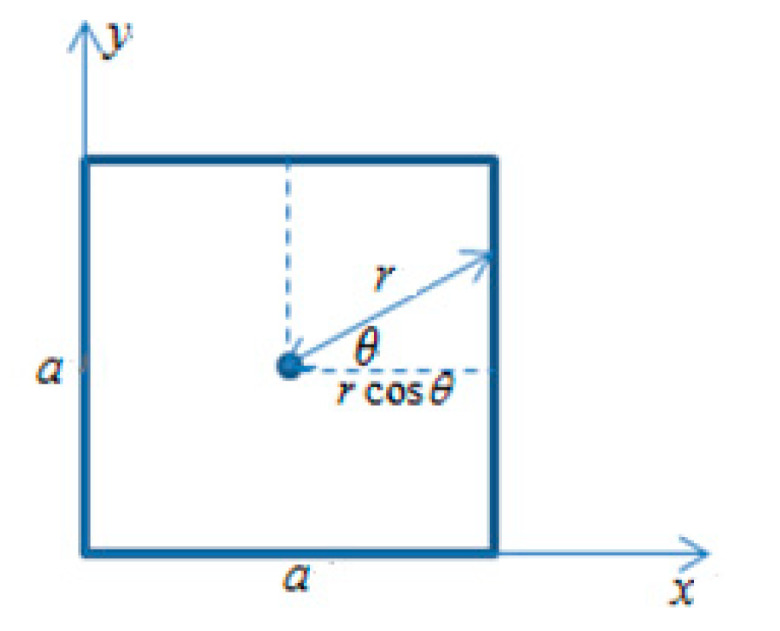
The schematic diagram of integration area.

**Figure 9 sensors-23-04050-f009:**
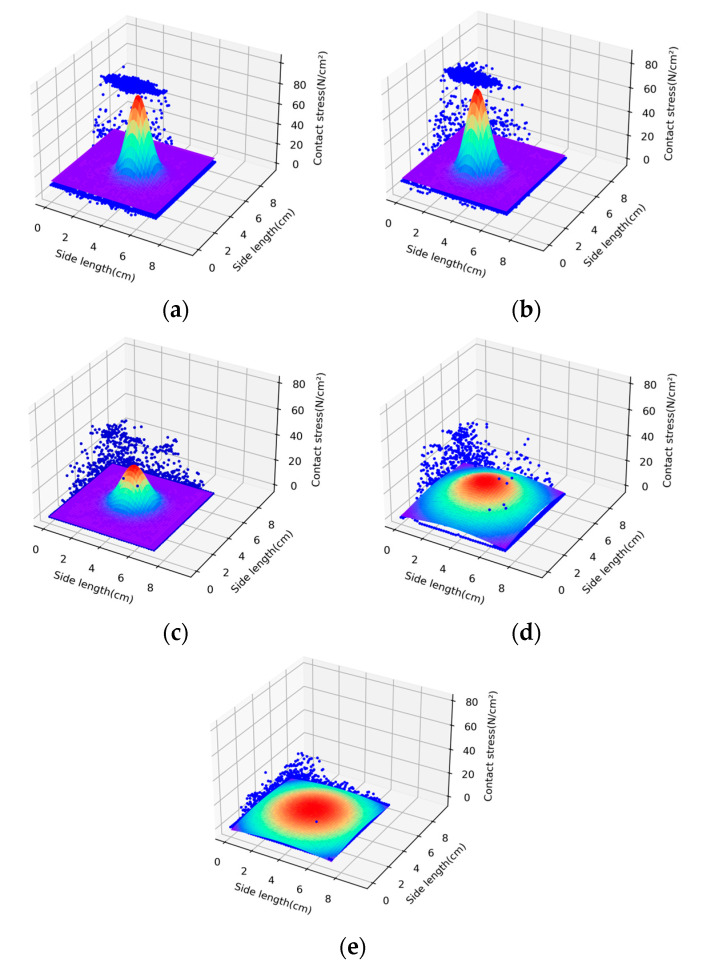
Comparison of 3D results of glass sheet with different aspect ratios under concentrated load: (**a**) aspect ratio of 18.75; (**b**) aspect ratio of 15; (**c**) aspect ratio of 9.375; (**d**) aspect ratio of 7.5; and (**e**) aspect ratio of 6.25.

**Figure 10 sensors-23-04050-f010:**
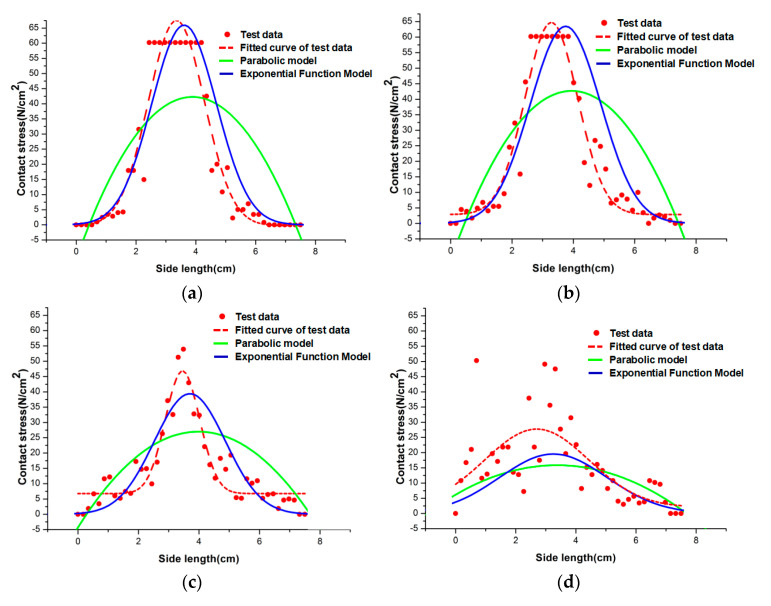
Comparison of contact stress values and experimental values of different models: (**a**) aspect ratio of 18.75 (λ = 18.75); (**b**) aspect ratio of 15 (λ = 15); (**c**) aspect ratio of 9.375 (λ = 9.375); (**d**) aspect ratio of 7.5 (λ = 7.5); and (**e**) aspect ratio of 6.25 (λ = 6.25).

**Figure 11 sensors-23-04050-f011:**
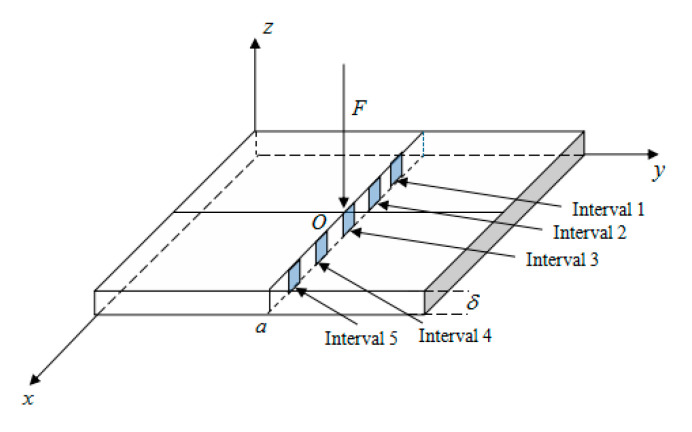
The interval selection diagram.

**Table 1 sensors-23-04050-t001:** Combined force value (N) and error of thin plate test results with different aspect ratios.

Test Load/N	λ = 18.75		λ = 15		λ = 9.375		λ = 7.5		λ = 6.25	
	Measurement Average Value/N	Error	Measurement Average Value/N	Error	Measurement Average Value/N	Error	Measurement Average Value/N	Error	Measurement Average Value/N	Error
250	142.68	−42.9%	264.16	5.7%	244.3	−2.3%	230.52	−7.80%	263.69	5.50%
500	399.83	−20.0%	523.02	4.6%	523.83	4.8%	505.66	1.10%	541.31	8.30%
750	729.49	−2.7%	750.7	0.1%	803.86	7.2%	778.93	3.90%	803	7.10%
1000	1085.28	8.5%	949.79	−5.0%	1069.65	7.0%	1044.14	4.40%	1049.04	4.90%
1250	1469.82	17.6%	1068.52	−14.5%	1333.97	6.7%	1295.76	3.70%	1285.7	2.90%

**Table 2 sensors-23-04050-t002:** Error comparison between different models and experimental stress values (N/cm^2^) (*λ* = 18.75).

Contact Stress Value	Interval 1	Interval 2	Interval 3	Interval 4	Interval 5	Percentage Error
Test value	12.66	25.75	60.19	60.19	27.43	
Exponential function model	11.97	23.44	56.43	57.76	26.33	3.80%
Parabolic model	11.72	29.67	40.75	41.44	32.91	21.20%
Linear model	17.78	17.78	17.78	17.78	17.78	49.49%

**Table 3 sensors-23-04050-t003:** Error comparison between different models and experimental stress values (N/cm^2^) (*λ* = 15).

Contact Stress Value	Interval 1	Interval 2	Interval 3	Interval 4	Interval 5	Percentage Error
Test value	11.49	26.12	60.02	42.46	30.01	
Exponential function model	11.07	24.13	56.42	38.1	27.62	3.40%
Parabolic model	11.31	30.68	40.97	42.16	34.25	7.10%
Linear model	17.78	17.78	17.78	17.78	17.78	51.18%

**Table 4 sensors-23-04050-t004:** Error comparison between different models and experimental stress values (N/cm^2^) (*λ* = 9.375).

Contact Stress Value	Interval 1	Interval 2	Interval 3	Interval 4	Interval 5	Percentage Error
Test value	9.92	21.02	23.35	29.34	23.36	
Exponential function model	14.25	22.66	25.99	29.75	24.56	3.20%
Parabolic model	8.86	19.97	25.89	26.7	22.38	7.95%
Linear model	17.78	17.78	17.78	17.78	17.78	36.35%

**Table 5 sensors-23-04050-t005:** Error comparison between different models and experimental stress values (N/cm^2^) (*λ* = 7.5).

Contact Stress Value	Interval 1	Interval 2	Interval 3	Interval 4	Interval 5	Percentage Error
Test value	18.69	22.72	27.2	22.08	22.08	
Exponential function model	18.98	22.43	24.51	24.74	23.09	4.60%
Parabolic model	10.87	14.47	15.85	15.01	11.96	39.54%
Linear model	17.78	17.78	17.78	17.78	17.78	20.03%

**Table 6 sensors-23-04050-t006:** Error comparison between different models and experimental stress values (N/cm^2^) (*λ* = 6.25).

Contact Stress Value	Interval 1	Interval 2	Interval 3	Interval 4	Interval 5	Percentage Error
Test value	21.86	23.49	24.72	24.31	22.88	
Exponential function model	21.72	22.57	23.03	23.08	22.72	4.80%
Parabolic model	8.87	14.05	16.91	17.45	15.68	38.18%
Linear model	17.78	17.78	17.78	17.78	17.78	24.03%

## Data Availability

All data generated or analyzed during this study are included in this published article.

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
