# Peer review of "Nonlinear Modeling of Contact Stress Distribution in Thin Plate Substrates Subjected to Aspect Ratio"

_sensors, 2023, doi:10.3390/s23084050_

Round 1
Reviewer 1 Report
Comments to Authors
The manuscript is well written with just sufficient novelty. However, its organization and presentation must be improved. Various figures like 1 with 2, 3 with 5 and 6 with 7 can be combined. Also, figures 11-15 can be combined into a single figure. Furthermore, figures 4, 8, 9 and 16 can be added in supporting information.
Author Response
Thank you sincerely for careful reviewing of our manuscript.Please see the attachment.

Reviewer 2 Report
I ask the authors to carefully read the attached concerns and make minor modifications to enhance the presentation of their paper.
1.The Introduction part did not give a clear review on the topic. It is suggested the Introduction part should be further revised in terms of contents and logic.
2.were the boundary conditions and effects considered?
3.The abstract section should be extended with a few words via the main finding and advantages of the methodology?.
4.How to verify the accuracy and correctness of the developed formulation and experimental set up?
5.I strongly recommend the authors to add one paragraph discussing the difference between their work and the previously performed studies in literature. In other words, what is the novelty of this work?
Author Response

(The authors gave the same response as above.)

Reviewer 3 Report
The scope of research presented in the paper is interesting. There is no justification for the analysis undertaken using glass samples, however, the authors inform about further research. There are editing errors, please pay attention to them. Very interesting job. Congratulations to the authors
Author Response

(The authors gave the same response as above.)

Reviewer 4 Report
Kindly see the attached Files.

Author Response

(The authors gave the same response as above.)

Round 2
Reviewer 1 Report
The manuscript can be accepted for publication.
Author Response
Thank you sincerely for careful reviewing of our manuscript.Checked and improved the language and style of the text.

Reviewer 4 Report
There are still some issues in the manuscript; kindly revise it before publication.
1. 0 MPa means the absence of pressure or stress, i.e., the nonexistence or absence of a specified thing, here in, i.e., stress. Kindly rewrite and change it to 0.01 MPa. Rewrite the statement in the following manner.
a. The sensors can be fabricated in various dimensions and configurations, exhibiting a stress measurement capacity ranging from 0.1 to 175 MPa.
2. Kindly also provide the side view of (Glass Slabs) Figure 2, i.e., showing the width variation, as one can’t evaluate the thickness proof from top view of glass slab images.
3. Point 14 in previous comments, Provide the reference with the statement.
4. Page 2, line 54: Provide reference of this “Design of Power Transmission Line Foundations 53 (DL/T5219-2005) and the Code for the Design of Building Foundations (GB50007-2002)” to display using 2.5 aspect ratio.
5. Page 9, 10,12. Lines and equations aren’t consistent; kindly recheck the final manuscript before submitting it.
6. Page 9, line 372: There is some issue with this statement.
a. The general aspect ratio factor k is taken to be between λ/5 and 2λ/15 .
What does 2λ/15 mean,
7. Figure 9 have again major flaws. The X and y axis scale must be consistent . the X-axis scale starts from 0 to 6, however Figure represents a negative value of initiating from -0.3 to 7.7. Kindly provide Figures with accurate and consistent scaling from 0 to 6 or 0 to 8 or -2 to 8. Similarly rescale Y-axis also from 0 to 100, to prove a consistency and comparison.There are still some issues in the manuscript; kindly revise it before publication.
Author Response

(The authors gave the same response as above.)
